# Materials Selection and Construction Development for Ensuring the Availability and Durability of the Molten Hydroxide Electrolyte Direct Carbon Fuel Cell (MH-MCFC)

**DOI:** 10.3390/ma13204659

**Published:** 2020-10-19

**Authors:** Andrzej Kacprzak, Renata Włodarczyk

**Affiliations:** Department of Advanced Energy Technologies, Faculty of Infrastructure and Environment, Czestochowa University of Technology, Dabrowskiego 69, 42-201 Czestochowa, Poland; akacprzak@is.pcz.pl

**Keywords:** direct carbon fuel cell, molten hydroxide electrolyte, corrosion, carbon steel, stainless steel, nickel

## Abstract

The molten hydroxide electrolyte Direct Carbon Fuel Cell (MH-DCFC) is a promising type of DCFC due to its advantages, such as high ionic conductivity, higher electrochemical activity of carbon (higher anodic oxidation rate and lower overpotentials) and high efficiency of carbon oxidation due to lower operating temperature (the dominant product of carbon oxidation is CO_2_ vs. CO). Accordingly, the MH-DCFC can be operated at lower temperatures (roughly 673–873 K), and thus cheaper materials can be used to manufacture the cell. Nonetheless, MH-DCFCs are still under development due to several fundamental and technological challenges such as corrosion problems. Selection of materials and development of a structure that ensures adequate availability and durability of the cell is crucial for the optimization of the MH-DCFC performance and the further development of that technology. This article presents the operating characteristics of the MH-DCFC made of different construction materials, such as carbon steel, stainless steel, and nickel and its alloys. Nickel and its alloys have proven to be the best materials for the construction of individual elements of the fuel cell. Inconel alloy 600 was a good catalytic material for cathodes with good corrosion resistance.

## 1. Introduction

The direct carbon fuel cell (DCFC) is a power generation device converting the chemical energy of carbon directly into electricity by electrochemical oxidation of the fuel. The basic structure of DCFC is similar to other cell types, such as Molten Carbonate Fuel Cell (MCFC) or Solid Oxide Fuel Cell (SOFC). However, in the DCFC, solid carbonaceous fuels (e.g., hard coals, brown coals, charred biomass, active carbons, carbon black, graphite, coke, etc.) are used and directly oxidized at the anode surface, and no gaseous fuels (e.g., H_2_ or CO) are required, as in the case of MCFC or SOFC.

Compared to other technologies, the solid carbon-fueled fuel cells have several unique features and advantages [1], offering higher thermodynamic efficiency and lower emission of carbon dioxide per unit of the generated electricity. Furthermore, the fuel for the DCFC does not require any sophisticated preparation, since the solid carbon can be easily obtained from various resources [1,2] such as coal, petroleum coke, charred biomass (e.g., grass, woods, nut shells, corn husks), or even organic garbage. There are four basic types of direct carbon fuel cells under development, which generally differ with respect to electrolyte types, which can be either molten carbonates [3,4], solid oxygen ion conducting ceramics [5,6,7,8], or aqueous [9] or molten hydroxides [10]. Composite electrolytes (so-called hybrid electrolytes) are also widely used in DCFC prototypes [11,12,13].

The molten hydroxide Direct Carbon Fuel Cell (MH-DCFC) is considered to be the most promising type of DCFC, due to its advantages [10], such as high ionic conductivity, the higher electrochemical activity of carbon (higher anodic oxidation rate and lower overpotentials) and higher efficiency of carbon oxidation due to the lower operating temperature (the dominant product of carbon oxidation is CO_2_ vs. CO). Accordingly, the MH-DCFC can be operated at lower temperatures (roughly 673–873 K), and thus cheaper materials can be used to manufacture the cell components.

MH-DCFCs are still under development, due to several fundamental and technological challenges, such as improvement of anode design in order to increase solid fuel reactive sites and reduce the cell cost, as well as maintaining cell operation at low temperatures, investigating carbon fuel characteristics and the required level of fuel pre-processing to maximize reactivity, providing a method for supplying solid carbonaceous fuels to the electrode/electrolyte interface, understanding the long-term effects of residual impurities in carbon-based fuels, technology scale-up, etc. Due to the strong corrosive behavior of the molten hydroxide electrolyte, the selection of materials resistant to that environment is also very important.

The corrosion processes of the cell elements accompanying the processes taking place on the MH-DCFC electrodes reduce the main electrical parameters, such as current and power density (both activation overpotential and ohmic polarization are increased). Moreover, substances generated as a result of these corrosive processes may contaminate the electrolyte and the surface of the carbon particles, which may cause an undesirable reduction in the redox reactions rate, resulting in an increase in the activation overpotential and concentration polarization.

The chemical reactions in the molten hydroxides cause corrosive processes in metals and alloys that are in contact with the electrolyte. From the point of view of the electrochemical corrosion, it is important to determine the durability of metals and their alloys due to the oxidizing properties of the corrosive environment. In the case of molten hydroxides, the corrosive effect is primarily determined by the type of hydroxide (NaOH, KOH, LiOH), temperature and the ability to dissolve oxygen.

Table 1 summarizes the corrosion rates of selected, generally available materials in a molten NaOH environment. The results relate to corrosion tests carried out at the operating temperature of the MH-DCFC (673 ÷ 773 K) and higher, i.e., from 853 K to 973 K. The tests were carried out by immersing the samples in molten hydroxide, and then measuring their weight changes over time. The corrosion rate was determined in mm year^−1^.

Analysis of the data contained in Table 1 indicates that nickel and its alloys—Monel 600, Monel 500, Inconel—show the highest corrosion resistance. Nickel-based alloys (3% Ni-Fe alloy) are also characterized by moderate corrosion resistance. It can also be seen that the corrosion rate increases as the temperature increases. Under the examined conditions, 301SS steel turned out to be attractive, both in economic terms as well as in terms of corrosion resistance. Austenitic steels, despite expensive alloying additions, do not show satisfactory corrosion resistance in molten NaOH. Corrosion rates comparable to those of austenitic steels were obtained for copper. At a temperature of approximately 773 K, the corrosion rate of copper was 17.2 mm year^−1^, and the corrosion rate of austenitic steel ranged from 11.8 to 24.4 mm year^−1^.

Considering the data summarized in Table 1, it appears that a number of construction materials can be used to manufacture the elements of direct carbon fuel cells with anode, cathode, electrolyte container, which are used in molten hydroxide electrolyte. Metals and alloys commonly used in MH-DCFC include carbon steel, stainless steel, nickel and high-nickel alloys [10]. All of these alloys have been selected, among others, to investigate directly in order to determine the material most suitable for target cell components construction in MH-DCFC [17].

Carbon steel (iron) is the predominant construction material used in the first generations of MH-DCFC [18,19,20]; therefore, it was used by the authors to build the first prototype of the fuel cell [17].

Austenitic stainless steel (SS) 300 series (i.e., SS 304L, SS 316L) was chosen to build the second prototype of MH-DCFC. 304 and 316 stainless steels are among the most commonly used types of stainless-steel materials in many applications. Type 304L and 316L SS are extra low-carbon variations of type 304 and 316 SS (e.g., max. carbon content for 304 SS is 0.08% and for the 304L SS max. carbon level is 0.030%). All other element ranges are essentially the same (the nickel range for 304 is 8.00–10.50%, and for 304L it is 8.00–12.00%). The lower carbon SS grades (304L and 316L) overcome the risk of intercrystalline corrosion. This can take place if the steel is held in a temperature range of 723 K to 1123 K for periods of several minutes, depending on the temperature, and is subsequently exposed to aggressive corrosive environments. Corrosion then takes place along the grain boundaries. If the carbon content is below 0.030%, then this intercrystalline corrosion does not take place following exposure to such temperatures. In general, 316L alloy is more corrosion resistant than type 304 stainless steels [21].

The third MH-DCFC prototype was manufactured from nickel and high-nickel alloys. Commercially pure nickel grades (Nickel 200 and 201) are used in a wide range of fields, such as in the aerospace and chemical industries (including hydroxide services) due to their high corrosion resistance. Nickel 201 is better suited in situations where high temperatures are applied, due to its lower carbon content [22,23]. However, nickel is rapidly oxidized in situ to NiO, which is a p-type semiconductor. The high corrosion resistance of Ni-based alloy Inconel 600 in molten alkaline solution of NaOH was attributed by Tran et al. [24] to the passive film of Ni–Cr spinel-type oxide which protects and prevents the leaching of Cr as chromite and chromate from the alloy.

The present work illustrates the construction and performance of a DCFC with hydroxide electrolyte. Three generations of MH-DCFC prototypes were built and tested in order to determine the influence of the molten hydroxides environment on the cell performance characteristics and durability of used materials such as carbon steel, stainless steels or nickel and its alloys.

## 2. Materials and Methods

### 2.1. MH-DCFCs Designs and Construction Details

#### 2.1.1. Model I

St3SU type carbon steel was selected for the construction of Model I of the MH-DCFC. The descriptions of the operation of one of the first carbon fuel cell prototypes, constructed in 1896 by Jacques, show that carbon steel is a cheap and readily available material showing good corrosion resistance in molten hydroxides at a temperature of 723 K [19,20]. Zecevic et al. [25] also confirmed the possibility of using carbon steel (additionally doped with titanium) for the construction of this fuel cell type. The addition of 1–2% Ti to carbon steel creates a layer of titanium oxide on the surface of the material, which is a degenerate semiconductor characterized by stable electrical conductivity and good corrosion resistance. This material also has very good catalytic properties towards oxygen, which allows it to be used in the construction of the cell’s cathode.

Model I of MH-DCFC was constructed based on the prototype constructed in 1896 by Jacques, but the distinguishing feature of Model I was the possibility of using granulated carbon fuel in the anode (Jacques used a carbon rod as a fuel in his prototype). During the experiments, the influence of molten alkaline environment on the St3SU type carbon steel was determined, whose elemental composition is summarized in Table 2.

The design scheme and a photo of Model I MH-DCFC are shown in Figure 1. The main part of the cell was a crucible (pot) made of carbon steel, in which molten electrolyte (NaOH) was contained. The steel pot also acted as the cell cathode. Electrons returning from the external circuit were transferred to the cathode through a special holder mounted on the outer part of the crucible. Air from a compressor (necessary for the electrochemical reaction) was distributed on the cathode surface by special pipe perforated at the bottom (see Figure 2). To maintain the electrolyte in the liquid phase the crucible was heated by an electric heater. Reduced heat loss was achieved by securing a prototype using mineral wool and sealing everything off in a ceramic casing.

The biochar particles were placed in a special “separator” in the form of a “basket” made of carbon steel. The side area of the “separator” was the basis for calculating the power and current densities. The use of such a structure made it possible to directly receive electrons from the biochar grains without using an additional current collector, adjusting the shape to the needs and prevent the fuel particles from entering into the electrolyte (at the same time allowing the electrolyte to freely penetrate into and out of the “separator”).

#### 2.1.2. Model II

Austenitic stainless steel AISI 300 series (304L, 316L, 316Ti), with increased corrosion resistance, was chosen to construct the second prototype of MH-DCFC, hereinafter referred to as “Model II”.

The alloy steels were covered with a layer of oxides consisting of iron and nickel—FeO, NiO, Fe_2_O_3_—and a smaller amount of mixed oxides such as NiCr_2_O_4_ after exposure to molten NaOH at a temperature of 673 K. The oxide layer on the steel surface, apart from metal oxides, is additionally formed by compounds such as Fe (OH)_3_ and FeOOH. The corrosion rate of alloy steels in molten NaOH significantly decreases in the temperature range 773–873 K. The protective properties of the oxide coating formed on the steel surface increase probably as a result of enriching it with NiO, Cr_3_O_4_, Cr_2_O_5_, CrO compounds. The aforementioned oxides form a layer of high density, evenly adhering to the surface and showing high resistance to chemical corrosion [15]. Based on literature reports [25] on the increased corrosion resistance of titanium doped steels, an alloy steel containing about 0.7% titanium (AISI 316Ti) was also used to build the cell cathode. The composition of the individual steels is described in Table 3.

The scheme and photograph of the MH-DCFC referred as Model II is shown in Figure 3. All the cell steel parts were manufactured from a corrosion-resistant stainless steel. The main electrolyte container was manufactured from two types of materials: Nickel 201 (in the lower part) and AISI 304L alloy (in the upper part) in order to provide sufficient corrosion resistance to liquid sodium hydroxide. The container was covered with ceramic band heaters providing heat for melting the electrolyte and maintaining the desired temperature level. The anode was made of AISI 304L steel and formed as by an especially designed tube with a diameter of 25 mm (see Figure 4a). Three windows were cut in the side of the anode tube. These windows on the outside were covered with a 50 μm mesh made of AISI 304L alloy steel, which was fixed to tube with two rings. The total geometric area of the mesh within the windows was the basis for calculating the current and power densities generated from the cell. The cathode, shown in Figure 4b, was manufactured from sintered stainless steels AISI 316Ti (bottom sparger, average pore size of 20 micrometers) and AISI 316L (main air pipe, outer diameter of 10 mm and wall thickness of 2 mm). Anode and cathode were mounted in the lid of the cell (the cathode is electrically isolated from the anode with a ceramic plug). Current collectors in the form of copper plates and wires were attached to the anode and cathode to assemble the external circuit.

#### 2.1.3. Model III

The last prototype, called “Model III”, was made of nickel and its alloys, and was characterized by the highest corrosion resistance. The analysis of the data contained in Table 1 shows that nickel and nickel alloys are characterized by very high corrosion resistance in molten NaOH at a temperature of 773 K. As a result of the contact of nickel with molten NaOH, NiO, Ni(OH)_2_ and γ-NiOOH phases were formed. The thickness of the oxide layer formed on the nickel surface depends on the conditions of diffusion of nickel ions through the oxide film and the rate of reaction between the nickel ions and the molten electrolyte. As a result of the described processes, compounds were formed, including, among others, Na_2_NiO_2_ and Na_5_NiO_4_. Due to the use of nickel, the forming oxide phases can slow down the electrochemical reactions occurs in the cell. Nickel oxidation changes the conductivity of the material: nickel is ferromagnetic, nickel oxide (NiO)-paramagnetic. Nickel alloys usually contain 45–65% nickel and are widely used in many industries due to their high corrosion resistance.

Nickel is one of the most resistant materials used to make individual components of the tested cell. Nickel is oxidized to NiO as a result of the oxidizing action of oxygen at high temperature. The thin, continuous layer of this oxide formed on the metal surface makes nickel resistant to the corrosive effects of aggressive environments. However, NiO is not a good electron conductor (stoichiometrically pure NiO has a specific electrical conductivity of only σ = 10^−14^ (Ω·cm)^−1^ [26]), so its electron conductivity must be increased to be a good current collector on the anode and cathode side. This can be achieved by doping the NiO semiconductor structure with cations of different valencies from that of the starting material. Doping the NiO oxide with Li^+^ ions causes the transition of Ni^2+^ ions into Ni^3+^ ions in an amount equivalent to the introduced Li^+^, which increases the number of gaps in the crystal lattice, and thus the electrical conductivity. The specific conductivity of the passive layer in the form of NiO oxide is 10^−4^ (Ω·cm)^−1^ in 1000 K, while after the doping process the conductivity increases to 10^−1^ (Ω·cm)^−1^ [27].

There are several methods for introducing lithium cations into the NiO layer. The most frequently used methods include electro-hydrothermal lithiation [28,29] and oxidation-lithiation used for the preparation of cathodes for MCFC cells carried out during the cell’s operation (in situ) or outside the cell (ex situ) [27,30]. In MCFC cells, due to the type of electrolyte, the Li^+^ ion donor (in situ method) is lithium carbonate (Li_2_CO_3_). In addition to lithium carbonate, the source of Li^+^ ions can also be lithium hydroxide (LiOH), lithium oxide (Li_2_O), lithium nitrate (LiNO_3_), or lithium chloride (LiCl).

A design scheme and photo of the Model III MH-DCFC are shown in Figure 5. This model was used to assess the effect of molten hydroxide electrolyte on the nickel and its alloys used for its construction. The types and elemental compositions of nickel alloys are presented in Table 4. Moreover, the study was carried out to determine the effect of an aggressive alkaline environment on the LiNiO_2_ coating prepared in the oxidation-lithiation process, which was responsible for improving the electrical conductivity of NiO formed under the cell’s operating conditions.

The main cell container was manufactured from the Nickel 201. The cathode of Model III (Figure 6) was a tube made of Inconel 600 alloy with an outer diameter of 42 mm (wall thickness 3 mm) and a length of 135 mm, welded from the bottom to the cell cover. The air was supplied to the lower part of the cathode by means of a pipe (outer diameter 6 mm and wall thickness 1 mm) placed centrally inside the main cathode tube. Holes with a diameter of 1 mm, drilled around the circumference of the cathode main tube, were used to distribute the air along the inside surface of the cathode tube. The anode (Figure 7) was made of a Nickel 201 alloy tube with an outer diameter of 19.1 mm and a wall thickness of 1.65 mm. In the side part of the anode, 74 holes, each 6 mm in diameter, were drilled, through which the electrolyte could flow into the anode chamber. Inside the tube, a coiled nickel mesh (mesh size 63 μm) was placed, the task of which was to stop fuel particles from entering into the electrolyte and to receive electrons generated in the electrochemical oxidation of carbon. For the calculation of the current and power densities, the geometric side surface of the nickel mesh was taken into account.

The components of the Model III MH-DCFC—in particular, the anode and cathode materials—were subjected to simultaneous oxidation-lithiation process. Doping cation-defective p-type nickel oxide (NiO) with lower valence cations (Li^+^) makes them highly conductive, which is a requirement for high-performance cathode and anode materials. The lithiated NiO was made by in situ oxidizing and lithium-doping Ni-base cathode and anode nickel alloys. The source of Li^+^ was lithium hydroxide monohydrate (LiOH∙H_2_O; melting point: 743 K). Shortly after the salt had been melted, air was introduced into the system (0.2 dm^3^ min^−1^) in order to accelerate the oxidation of nickel to NiO, and then lithium ions could be incorporated into the inner surface of NiO film. The process was carried out at 873 K for 24 h. Afterwards, anode and cathode materials were slowly cooled to room temperature, and then were washed several times in distilled water and HCl solution (0.05 M) to remove and neutralize the residue of lithium hydroxide. After washing, the cell components were dried in a convection oven for 2 h. A depiction of the anode and cathode elements after the lithiation process is shown in Figure 8.

Additionally, in order to determine the influence of the molten hydroxide environment on the durability of the produced LiNiO_2_ layer, a special ring was placed inside the cathode chamber (see Figure 9). This ring was weighed before and after each test, and then the fragment was cut off, and XRD analysis of the surface was performed in order to determine whether, during the lithiation process, lithium inclusions were formed in the NiO structure in the form of LiNiO_2_ compounds.

### 2.2. XRD and Microstructural Analysis

Selected materials used for construction of the cell were subjected to X-ray Diffraction (XRD). XRD is a non-destructive technique for material analysis, including for semiconductors that include passive layers formed when exposed to molten hydroxides of cell surface materials. X-ray diffractometry is a technique used in crystallography in which a diffraction image, generated by X-ray diffraction through a spatial network of atoms in a crystal, is recorded and analyzed to determine the structure of the network. It is a technique used to learn about the materials and molecular structures of the substances studied. X-ray diffraction measurements enabled phase identification of the layer on a metallic substrate. The materials were analyzed using a Seifert 3003 T-T diffractometer with a cobalt anode, λ_K_αCo_ = 0.179 nm.

Microstructure analysis was carried out using a scanning electron microscope (SEM); Philips XL30/LaB6.

### 2.3. Weight Loss Analysis

The corrosion rate was studied with the use of the weight loss technique. Measurements on test materials were conducted under the same conditions and the results were compared. Weights were determined with a precision of 0.0001 g on a RADWAG AS 220/X digital analytical balance.

### 2.4. Characterization of Carbon Fuels

Two different carbon fuels with different characteristics were used—graphite rod and commercial biochar. The example fuel samples are shown in Figure 10, while the main properties of the fuels are shown in Table 5. The particle size of the biochar fuel samples used in the experiments was in the range of 0.18–0.25 mm.

### 2.5. MH-DCFC Performance Tests

The special test setup was developed to allow the measurement of the cell’s current and voltage as well as preparation of the necessary characteristics of the MH-DCFC operation, on the basis of which the cell performance was evaluated. The scheme and view of the measurement system is shown in Figure 11.

The electrolyte temperature was determined by a K-type thermocouple (NiCr-NiAl) and was maintained at the desired value by an electronic temperature controller. The Tektronix DMM 4040 digital multimeter was used to measure the open circuit voltage of the fuel cell. To determine the cell current intensity at various loads, an external resistance setup MDR-93/2-52 was used and connected to the cell circuit thus providing the possibility to adjust the electrical resistance of the external circuit (in the range 0.1–10,000 Ω). The data acquisition module Advantech USB-4711A was used for the measurement of the cell voltage and the decrease of the voltage on an external resistor. The obtained average voltage values (at the cell terminals and the external resistance terminals) were used to calculate the current (and power) using the equations:(1)$$I=\frac{U_{z}}{R_{z}}$$
(2)P=U·I
where:*I*—current in the circuit, [A],*U_z_*—voltage drop across the resistor, [V],*R_z_*—resistances of resistor, [Ω],*P*—cell power, [W],*U*—voltage at the cell terminals, [V].

The module was connected to a personal computer (PC) where the data was displayed and stored. The amount of air fed into the cell was controlled by a thermal mass flow controller (Brooks 4850) with a Local Operator Interface (LOI) to view, control and configure the control device. It was possible to adjust the gas flow rate from 0.1 dm^3^ min^−1^. To 2 dm^3^ min^−1^. To attenuate short-term surge suppression and eliminate the effects of power grid interferences on the recorded data, the emergency standby backup power device PowerCom UPS BNT-1500AP, with a noise filter EMI/RFI, was also used during the experiments. The tool also acted as an ‘emergency power supply device’ for the data recording system in case of power failure.

At the beginning of each test, an appropriate amount of sodium hydroxide was put into the main cell container (Model I and II) or eutectic mixture of NaOH (90 mol%) with LiOH (10 mol%) in case of Model III and then heated up to the desired temperature. After the temperature level of 723 K was reached and the electrolyte was completely melted, both the cathode and the anode were slowly immersed into the electrolyte and the cell data (current, voltage, temperature, etc.) were recorded. The air to the fuel cell was fed from a compressor (pressure of 0.15 MPa) with controlled flow rate of 0.5 dm^3^ min^−1^.

## 3. Results

### 3.1. Model I

The variations of the cell voltage plotted against time for the current I = 0 A (open circuit voltage—OCV) are shown in Figure 12a, and the relationships between cell voltage, power density, and current density (performance characteristics; polarization curves) for a commercial biochar and graphite are plotted in Figure 12b. Performance characteristics were determined after 3 h of cell operation without external load.

As indicated by the results presented in Figure 12a, the OCV values increased slowly until they reached a stable plateau after roughly 1 h and then remained quite stable. The differences between the OCV values, particularly at the initial period of cell operation, were probably brought about by gradual wetting of the inner surface of the fuel (biochar particles and graphite electrode surface) by liquid molten electrolyte [31]. The OCV values measured by digital multimeter after 3 h of operation were 1.075 V and 0.852 V for biochar and graphite, respectively.

The voltage–current density characteristics (Figure 12b) show that the limiting current condition is reached nearly 7.77 mA cm^−2^ with biochar as a fuel and 4.2 mA cm^−2^ for graphite, while the maximum values of power density were respectively: 5.1 mW cm^−2^ and 2.5 mW cm^−2^.

The differences in cell operation, associated with a particular fuel, are probably the effect of the fuel specific surface area (and thus the reaction kinetics), or are brought about by so-called crystallographic disorders (crystal structure disorder). Compared with graphite, the structure of biochar is in a more disordered form. The reactivity of disordered carbon is much higher than that of the graphitic carbon because the disordered carbon contains many more edge sites as well as various types of defects that are more reactive than crystalline sites.

Polarization is caused by chemical and physical factors associated with various elements of the fuel cell. These factors limit the reaction processes when current is flowing. The current–voltage characteristics shape (Figure 12b) depends on many factors, but in general it can be seen that both curves were dominated by the ohmic polarization (middle region of the polarization curves) related to the resistances of the electrolyte and electrodes. In both cases, regions related to concentration polarization (end regions of the characteristics) are also noticeable. However, only graphite was characterized by a clear region associated with activation losses (occur at low currents), which may be related to its lower reactivity compared to biochar.

During the analysis of the fuel cell operation, progressive corrosion of carbon steel and the presence of corrosion products in the electrolyte were observed. The formation of non-conductive oxide layers on the surface of the crucible (cathode) and separator (anode) could have a major impact on the concentration polarization. Even if mass transport (supply of substrates and removal of products to and from the surface of the electrodes) was fast enough, as a result of the formation of oxide layers, it was difficult to transfer the electric charge. In addition, in the case of a solid graphite electrode, its surface was covered with a layer of corrosion products, which probably limited the transport of hydroxyl ions to reaction zones. The described processes probably had the greatest impact on the high value of the internal resistance of the cell, which in the case of graphite was 2.19 Ω, while for charcoal it was 1.69 Ω.

The second stage of the research was to carry out the long-term test for Model I. In this case, biochar was used as fuel (0.5 g). The cell worked under a load of 20 Ω, which corresponded to the maximum power obtained during determination of cell performance characteristics. The voltage and current values determined during the long-term experiment are plotted in Figure 13.

The fuel cell, after working for 27 h, reached a final voltage of about 0.09 V. It can also be seen that the current intensity, despite the progressive voltage drop, maintained a value in the range of 10 to 18 mA throughout the entire duration of the test. Strong corrosion of the cell construction materials was found after the test, which can be seen in the pictures in Figure 14. In addition, the electrolyte had a dark brown color, indicating the penetration of corrosion products into it. The production of non-conductive oxide layers on the cell elements, in particular on the crucible (cathode) and separator (anode) surfaces, could have an impact on the observed voltage and current fluctuations. Since the current did not decrease over time, as was the case for voltage, and after the experiment a biochar weight loss of only 0.12 g (24% compared to the initial mass) was found, it can be concluded that the voltage drop at the electrodes of the cell was caused by progressive corrosion.

Carbon steel, despite the positive results obtained by Jacques, quickly degraded in the environment of molten NaOH. The main corrosion product is iron oxide (Fe_2_O_3_), which is shown in the diffraction pattern recorded for the electrolyte tank fragment (cathode) after the cell had finished working (Figure 15). In the case of the separator, the visible black color corresponds to fragments of unreacted fuel (biochar).

Similar results were obtained by Zecevic at al. [25]. Carbon steel oxidizes very quickly and corrodes in the presence of oxygen contained in the electrolyte. The cathode surface becomes nonconductive after oxidation due to the formation of Fe_2_O_3_. On the other hand, the cathode, made of mild steel with addition of titanium, enables cell operation for over 540 h. With the use of mild steel doped with 2 wt% of titanium (Fe2Ti), the formation of a layer of oxide is caused on the material surface, which represents the degenerated semi-conductor characterized by a stable electrical conductivity and good corrosion resistance. This material also has very good chemical stability and promising catalytic properties for the oxygen reduction reaction [25].

### 3.2. Model II

The variations in OCV plotted against time, and the relationships between cell voltage, power density, and current density, are shown in Figure 16.

As in the case of Model I, the OCV slowly increased over time and reached a plateau after an hour (see Figure 16a). The voltage measured after 3 h of cell operation was 0.95 V for biochar and 0.6 V for graphite. The use of different materials for the construction of the anode and cathode was probably the reason for obtaining lower OCV voltage values compared to Model I (it is probably that the values of the potentials generated on the electrodes were different).

As can be seen in Figure 16b, Model II produced an increased performance, with maximum power densities of above 6.7 mW cm^−2^ with graphite rod serves as a fuel. For biochar, the maximum power densities were comparable with the first prototype and equaled to 4.8 mW cm^−2^.

The use of materials characterized by greater corrosion resistance in the environment of molten hydroxides in the construction of Model II improved the quality of the electrolyte, which could be visually assessed. This probably resulted in an increase in the obtained power density (especially in the case of graphite), as well as a lower internal resistance (1.54 Ω for graphite and 1.19 Ω for biochar).

Model II was also subjected to a long-term test in order to determine the stability of operation and the influence of the molten NaOH environment on individual cell components, and thus on the cell’s lifetime. In this case, biochar was also used as fuel (9.0 g), and the cell was operated under a load of 50 Ω. The voltage and current values determined during the long-term experiment are plotted in Figure 17.

The fuel cell operation was analyzed for 93 h, after which a voltage drop at the terminals below 0.1 V was noted. After the experiment, a biochar mass loss of 3.1 g (≈34% compared to the initial weight) was found. Due to the materials used, the cell worked three times longer than Model I; however, the progressive corrosion of individual cell elements negatively affected the operation of the device, which could be indicated by the observed voltage and current fluctuations. Figure 18 presents photos of the anode, cathode and electrolyte crucible after the long-term test.

The different colors of the layers that have formed on the surface of the cell elements after long-term testing allow to determine the type of oxides:red, brown—iron (III) oxide; Fe_2_O_3_,dark brown—iron (II) chromate; Fe_2_CrO_4_,yellow—iron (III) oxide monohydrate; Fe_2_O_3_ ∙ H_2_O, sodium chromate; Na_2_CrO_4_,black—nickel (II) oxide; NiO.

The collection of electric charges from biochar particles was done through a steel mesh. The steel mesh during the operation of the cell was covered with an oxide layer, which in addition to iron oxides also formed nickel and chromium oxides. These oxides show high electrical resistance, which could have worsened the electron charge collection from biochar particles, and this in turn would explain the lower operating parameters obtained for biochar compared to graphite, for which the charge was collected directly from the electrode surface. Like the Model I, oxide layers formed on the surface cell anodes and cathodes were probably the main reason for the observed voltage and current fluctuations.

Model II elements are made of 304L and 316L corrosion-resistant alloy steel (containing chromium and nickel—additives responsible for increasing the corrosion resistance of steel). The electrolyte tank in the case of Model II was built of two types of materials: in the lower part made of nickel, in the upper part made of 304L alloy steel. The heterogeneity of the material from which the crucible was made resulted in accelerated degradation of steel as a less noble material during the operation of the cell, under current conditions, at the Ni/304L contact point. At the interface of both materials, a micro-cell and steel (and especially the steel component, i.e., Fe) were formed, undergoing oxidation. The result of steel digestion was visible as a layer of rust in the upper part of the crucible (see Figure 18c).

The cathode was a 316Ti stainless steel tube with a porous air sparger made of sintered 316L steel powder. It can be seen that 316Ti steel was disintegrated and corrosion products were deposited on the surface of the tube. In the case of the air sparger made of 316L steel, a black oxide layer is visible, which is most likely NiO.

The XRD patterns (Figure 19) of the anode and cathode surface after long-term operation of the Model II shows that apart from the austenitic phase of iron (III) oxide (hematite; Fe_2_O_3_) and iron oxyhydroxide (lepidocrocite; γ-FeO[OH]), a solid residue of Na_2_CO_3_ was formed on the surface of the anode tube resulting from the carbonation of the electrolyte.

### 3.3. Model III

In the case of the MH-DCFC Model III, the electrolyte was a mixture of molten sodium and lithium hydroxides (molar ratio: 9:1). Lithium hydroxide was the donor of Li^+^ ions, which served the in situ lithiation process of the NiO layer during the cell operation. Figure 20 shows the OCV changes vs. time and the performance characteristics of the Model III fueled by biochar and graphite.

Using nickel and its alloys in the construction of Model III contributed to a significant increase in the most important electrical parameters, such as OCV, power and current densities, when biochar was used as a fuel. In turn, for graphite, the change in the cell structure and the materials used resulted in a slight decrease in the current and power density values, which could be caused by a change in the electrolyte composition (pure NaOH was used in Models I and II, while the electrolyte in Model III was the NaOH-LiOH mixture).

The results shown in Figure 20a indicate that, in the case of graphite, the cell reached a stable voltage after an hour, while for biochar, the voltage stabilization was achieved after 2 h. Finally, after 3 h of non-load operation, the measured OCV was equal to 1.05 V and 0.67 V for biochar and graphite, respectively. The estimated internal resistance was 0.39 Ω for graphite and 0.52 Ω for biochar. The obtained maximum power and current densities were, respectively, 5.3 mW cm^−2^ and 29.5 mA cm^−2^ for graphite and 33.5 mW cm^−2^ and 75.0 mA cm^−2^ for biochar.

The decrease in the internal resistance value and the increase in other electrical parameters (compared to Models I and II) were most likely related to the reduction of corrosion of the cell components (in particular the anode and cathode). Moreover, after each of the tests, no contamination of the electrolyte with corrosion products was observed.

Positive results of the preliminary tests of model III, similarly to Models I and II, were verified with the long-term test. The mass of used biochar was 2.6 g, while the external load of the cell was set to 1 Ω. The voltage and current values recorded during the long-term experiment are presented in Figure 21.

The fuel cell worked for more than 43 h, after which a voltage drop at the terminals to about 0.02 V was noted. After the experiment, it was found that the entire mass of biochar had reacted, which may explain the gradual decrease in current and voltage over time. No corrosion products were observed in the electrolyte after the test. Similarly, the construction materials did not show any signs of corrosion, and their appearance was identical to that after the lithiation process (cf. Figure 8). Due to the elimination of the problem of progressive corrosion of cell elements, the explanations required visible fluctuations in voltage and current during cell operation. The most probable cause of the occurring fluctuations was the decreasing amount of fuel in the separator and the associated loss of biochar particles in contact with the current collector (nickel mesh).

Molten hydroxide electrolyte DCFC manufactured from nickel was also examined by Guo el al. [32]. No significant changes in the value of recorded electrical parameters generated from the cell and no significant electrolyte degradation were found during 100 h of experimentation. The test performed for various operating temperatures of cells in the range of 643–823 K showed that the increase in temperature leads to the increased conductivity of both electrolyte and oxide layers formed on the surfaces of electrodes while cells’ power also increased [32].

In the case of Model III, electrodes were made of nickel (Nickel 201) and high-nickel alloy (Inconel 600). To observe surface and structural changes of materials exposed in molten NaOH, microstructure tests were performed. Figure 22 presents the microstructures of the tested materials after a long-term test. The structure of the Inconel 600 and Nickel 201 alloys indicates that a good adherent and protective oxide layer was formed. No pitting or similar material discontinuities can be observed. The bright areas in the pictures are products resulting from the reaction of the electrolyte with the material. XRD analysis (Figure 23) indicated that the main components of Inconel 600 material, regardless of the time the material was exposed in the electrolyte, were the NiO phase and the LiNiO phase. The presence of the lithium–nickel phase, even after 43 h of exposure, was satisfactory, because it means that the surface layer containing the LiNiO phase continuously protects and maintains its semiconductor properties by doping nickel oxide with lithium (which was the intention of the lithiation process).

As can be seen from the analysis of the data summarized in Table 6, the nickel lithiation process prevents corrosion. The weight increase of the sample after 77 h of cell operation may result from the strengthening of the nickel oxide layer on the material surface as well as the simultaneous incorporation of lithium ions in the NiO structure.

## 4. Conclusions

From the experimental results presented and briefly discussed in the present paper, the following conclusions can be formulated:Three MH-DCFCs with different configurations and construction materials were successfully tested at the laboratory scale. The obtained results clearly indicate that the effects of construction materials on the operation performance of the MH-DCFC are quite promising with respect to the potential application of the DCFC technology for large-scale power generation.The performance of the direct carbon fuel cell with molten hydroxide electrolyte can be significantly improved by optimizing the cell design and the electrodes materials.Carbon steel, despite the positive results obtained during the first MH-DCFC tests conducted by Jacques in 1986, quickly degrades in an environment of molten NaOH.AISI 300 series austenitic stainless steels are materials characterized by greater corrosion resistance in the environment of molten hydroxides. However, the obtained results indicate that these steels are not suitable for the construction of the MH-DCFC, due to their progressive corrosion processes, resulting in the reduction of the cell’s lifetime.Nickel and its alloys were proven to be the best materials for the construction of individual elements of the fuel cell. Inconel alloy 600 was a good catalytic material for the cathode with good corrosion resistance.The shape of the current–voltage characteristics depends on many factors, but in general it can be seen that those curves were dominated by the ohmic polarization related to the resistances of the electrolyte and electrodes. Therefore, it is very important to avoid corrosion of the cell’s construction materials as well as the leakage of corrosion products into the electrolyte.

## Figures and Tables

**Figure 1 materials-13-04659-f001:**
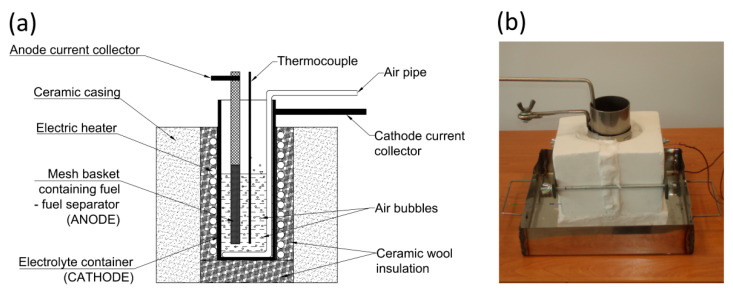
Design scheme (**a**) and photograph (**b**) of experimental Model I of MH-DCFC made of carbon steel.

**Figure 2 materials-13-04659-f002:**
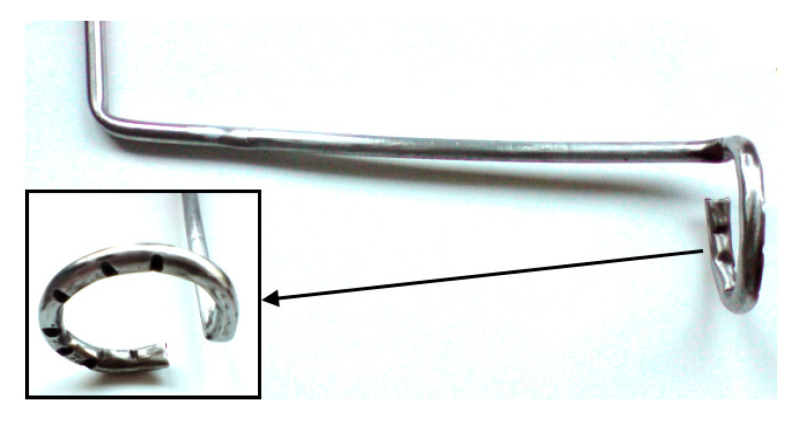
View of the construction of the air supply and distribution pipe made of carbon steel.

**Figure 3 materials-13-04659-f003:**
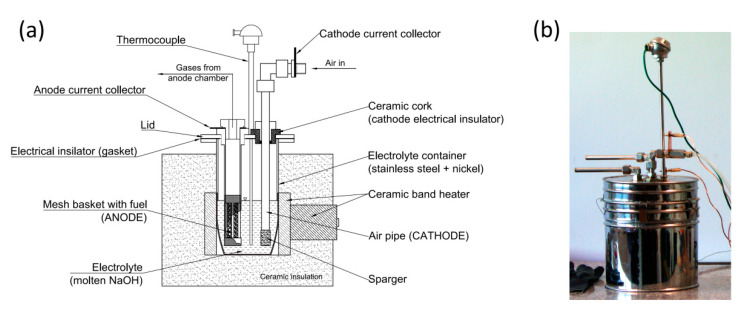
Design scheme (**a**) and photograph (**b**) of experimental model II of MH-DCFC made of stainless steel AISI 300 series.

**Figure 4 materials-13-04659-f004:**
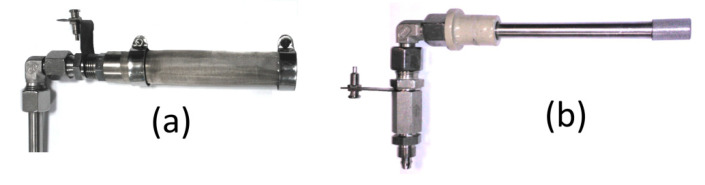
View of the anode (**a**) and cathode (**b**) of the Model II MH-DCFC.

**Figure 5 materials-13-04659-f005:**
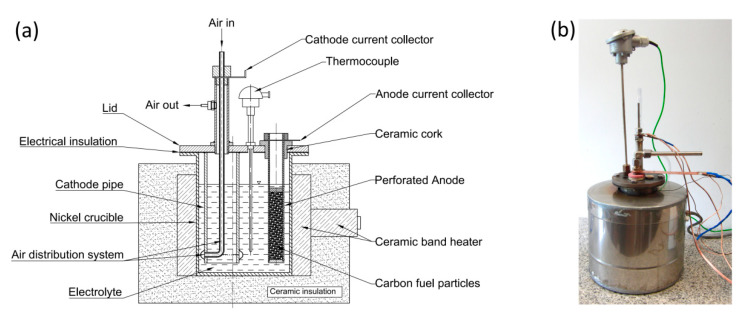
Design scheme (**a**) and photograph (**b**) of experimental Model III of MH-DCFC, made of nickel and high-nickel alloys.

**Figure 6 materials-13-04659-f006:**
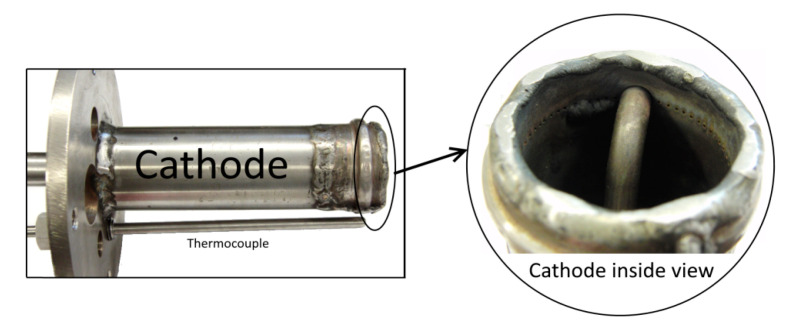
View of the Model III cathode construction before the lithiation process.

**Figure 7 materials-13-04659-f007:**
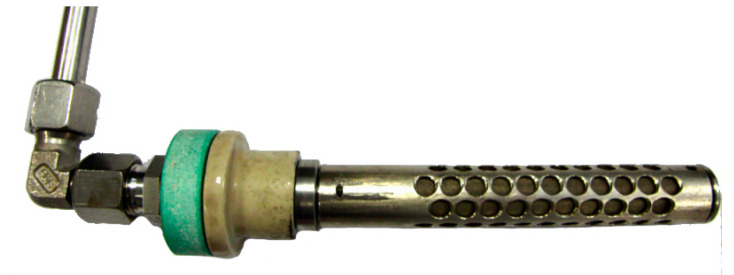
View of the Model III anode construction before the lithiation process.

**Figure 8 materials-13-04659-f008:**
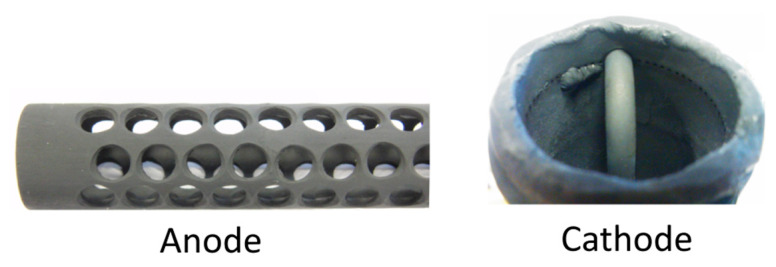
View of the anode and cathode elements after the lithiation process.

**Figure 9 materials-13-04659-f009:**
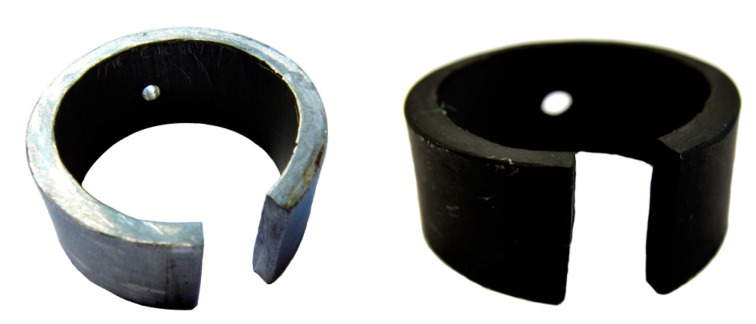
Depiction of the Inconel 600 alloy ring: before (**left**) and after the oxidation-lithiation process (**right**).

**Figure 10 materials-13-04659-f010:**
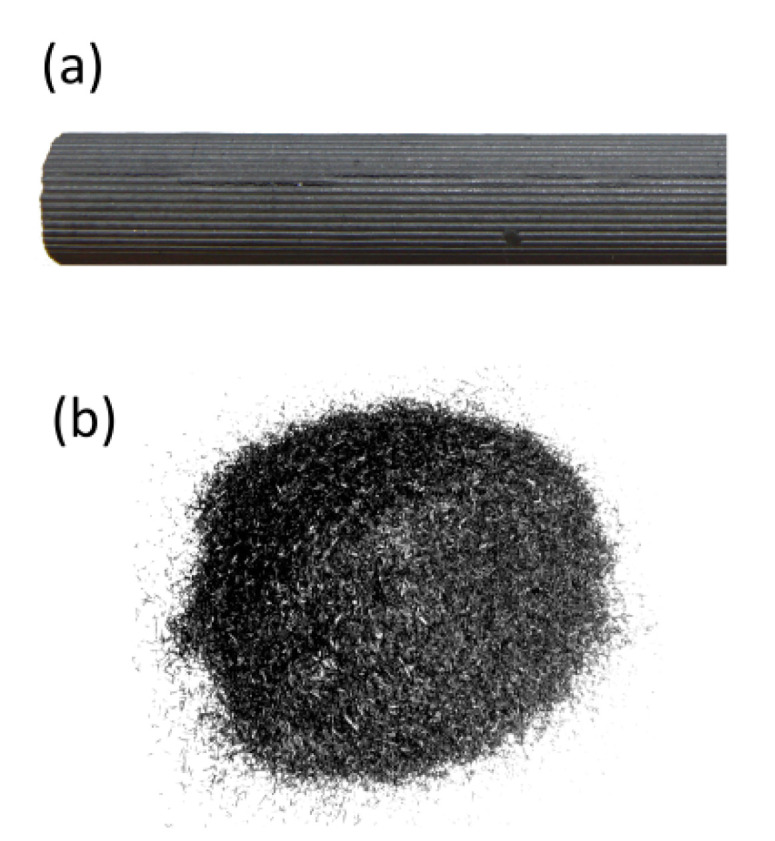
The example fuel samples used to supply the MH-DCFCs models: (**a**) graphite rod (diameter: 13 mm), (**b**) commercial biochar (particle size: 0.18–0.25 mm).

**Figure 11 materials-13-04659-f011:**
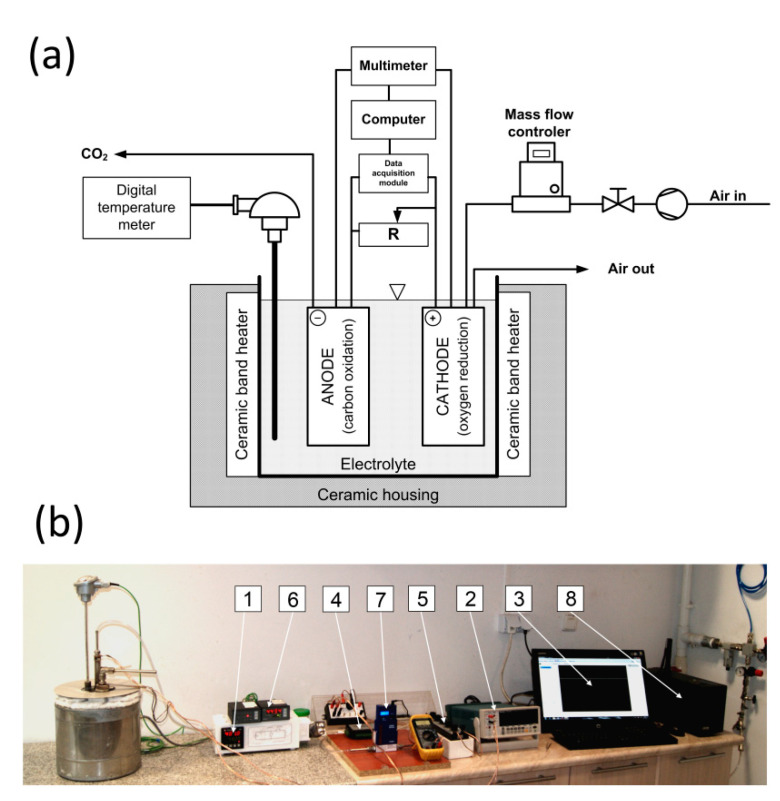
Schematic diagram (**a**) and depiction (**b**) of the MH-DCFC test setup. 1—electronic temperature controller (thermostat), 2—digital multimeter, 3—computer PC, 4—data acquisition module, 5—adjustable external resistor, 6—digital temperature display, 7—mass flow controller, 8—backup power device.

**Figure 12 materials-13-04659-f012:**
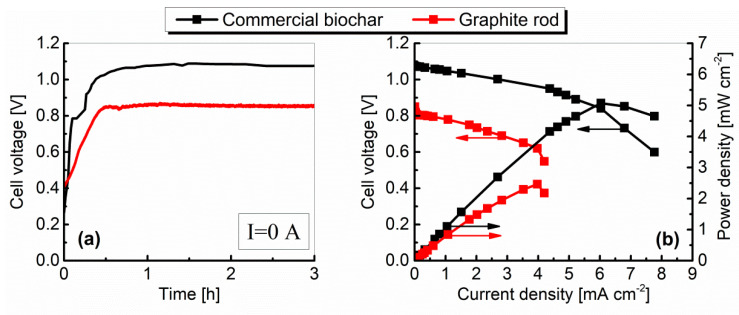
The variation of the fuel cell voltage vs. time (**a**) and the performance characteristics (**b**) of MH-DCFC Model I constructed from carbon steel.

**Figure 13 materials-13-04659-f013:**
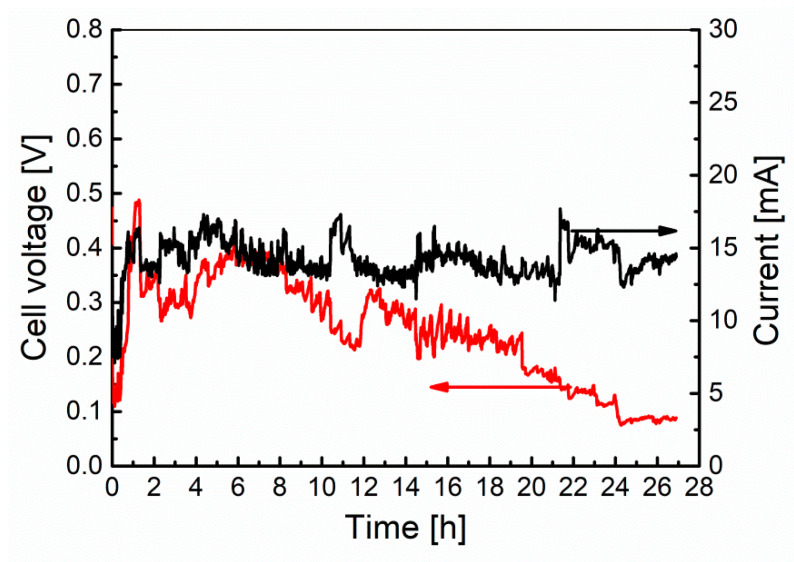
The variation of the fuel cell voltage and current vs. time during long-term experiment test of the Model I MH-DCFC.

**Figure 14 materials-13-04659-f014:**
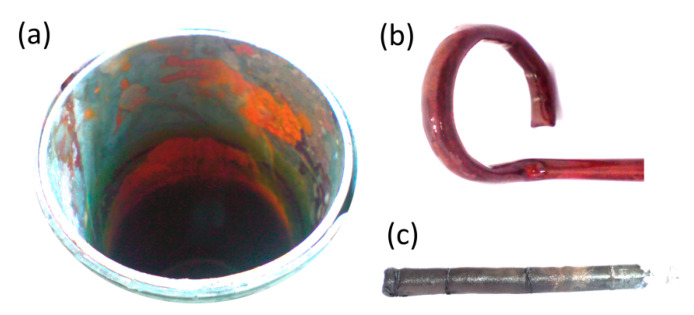
Images of individual construction parts of Model I after the long-term test: (**a**) crucible (cathode), (**b**) air supply pipe, (**c**) separator (anode).

**Figure 15 materials-13-04659-f015:**
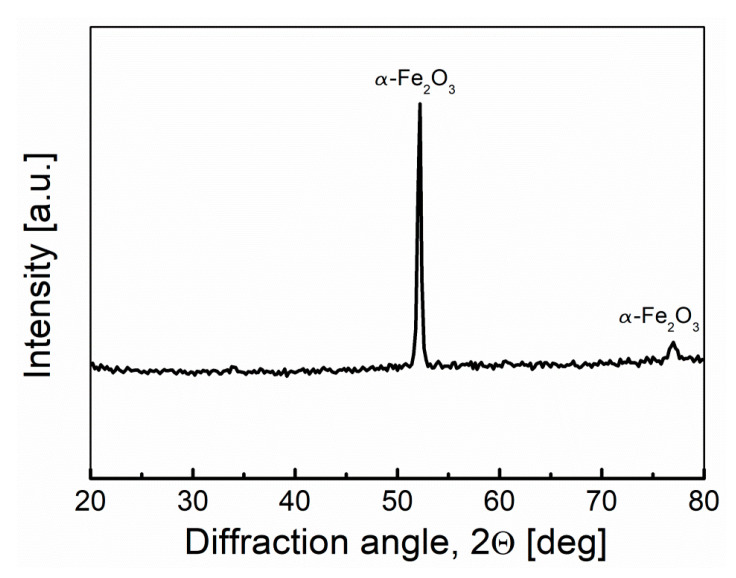
XRD pattern recorded for the electrolyte container surface—cathode of Model I MH-DCFC.

**Figure 16 materials-13-04659-f016:**
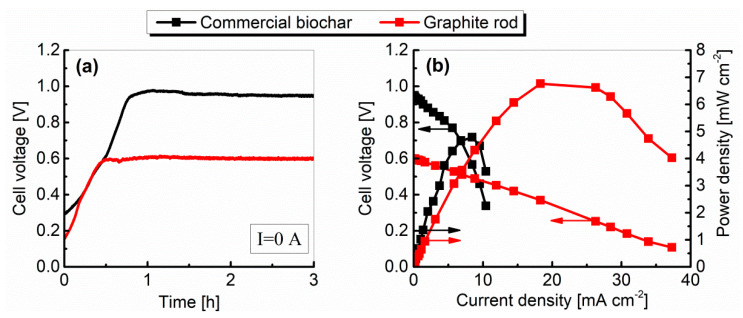
The variation of the fuel cell voltage vs. time (**a**) and performance characteristics (**b**) of MH-DCFC Model II constructed from austenitic stainless steel 300 series.

**Figure 17 materials-13-04659-f017:**
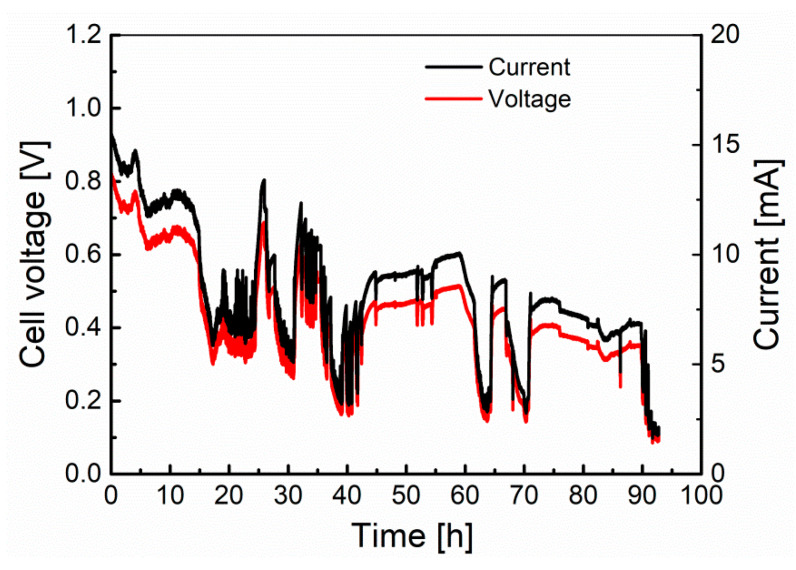
The variation of the fuel cell voltage and current vs. time during long-term experiment test of the Model II MH-DCFC.

**Figure 18 materials-13-04659-f018:**
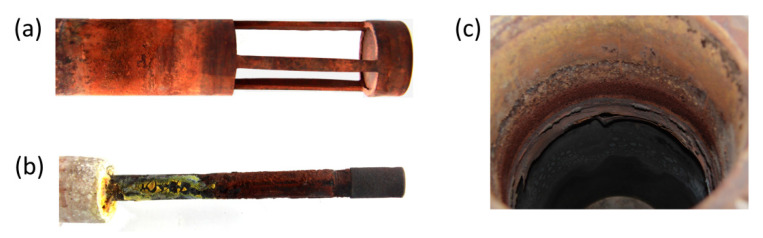
Images of individual construction parts of Model II after the long-term test: (**a**) anode, (**b**) cathode, (**c**) electrolyte crucible.

**Figure 19 materials-13-04659-f019:**
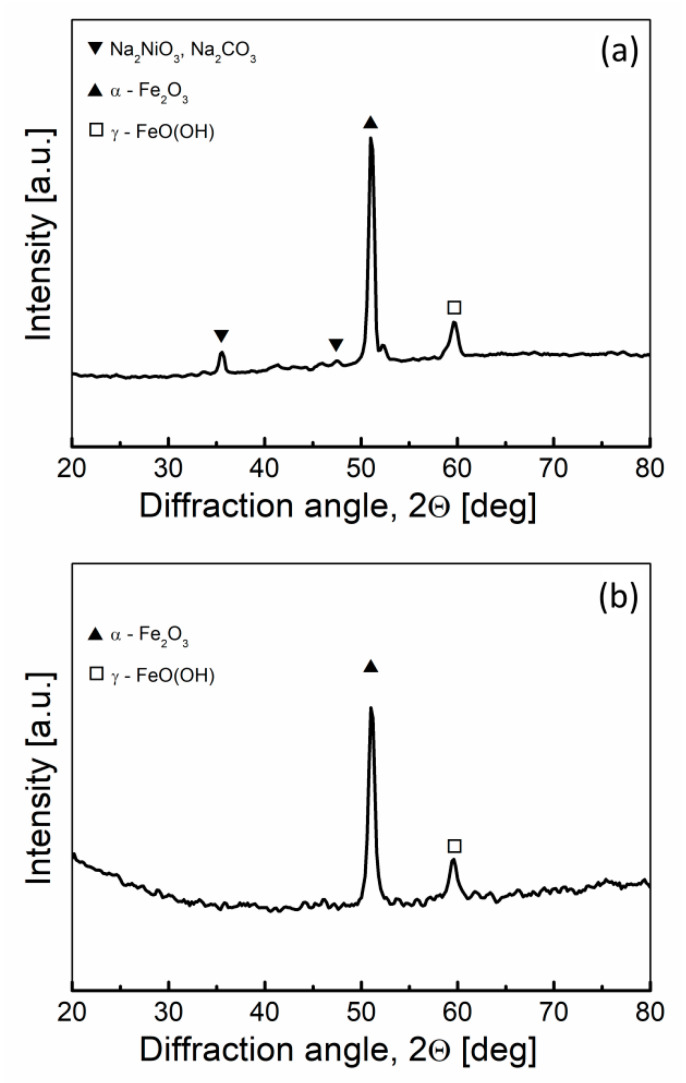
XRD patterns recorded for the elements of the Model II MH-DCFC: (**a**) anode, (**b**) cathode.

**Figure 20 materials-13-04659-f020:**
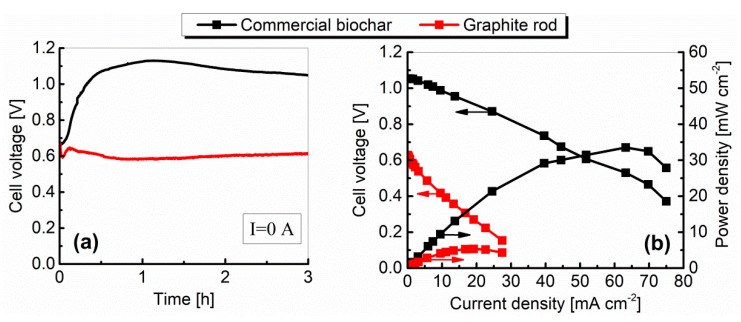
The variation of the fuel cell voltage vs. time (**a**) and performance characteristics (**b**) of the MH-DCFC model III constructed from nickel.

**Figure 21 materials-13-04659-f021:**
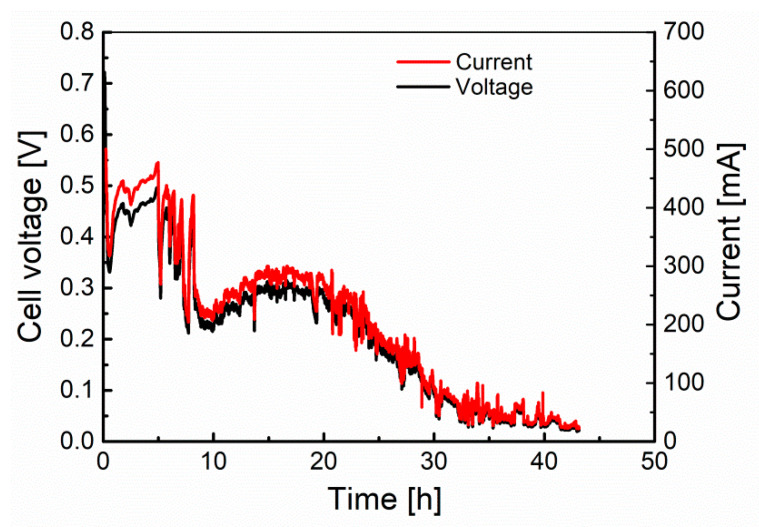
The variation of the fuel cell voltage and current vs. time during the long-term experiment test of the model III MH-DCFC.

**Figure 22 materials-13-04659-f022:**
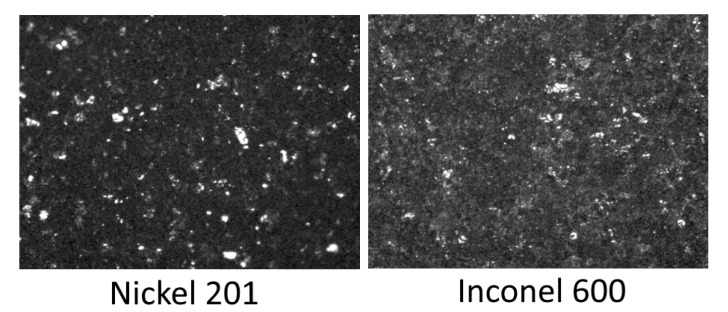
Scanning electron micrographs of the materials used to build the Model III after 43 h long-term test. Magnification: 100×.

**Figure 23 materials-13-04659-f023:**
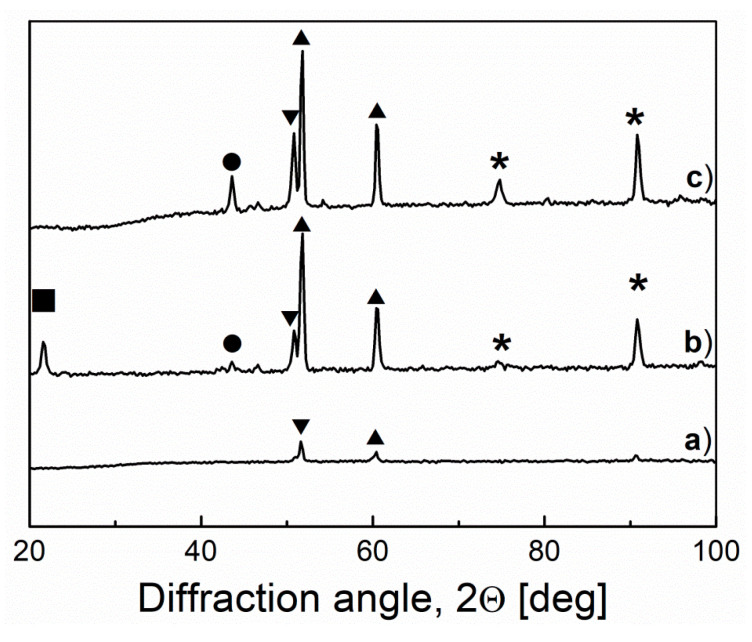
XRD pattern recorded for the raw Inconel 600 (**a**), Inconel 600 after lithiation process (**b**) and Inconel 600 after 43 h of cell work (**c**). (■) Li_0.4_Ni_0.6_O, (●) Li_0.05_Ni_0.95_O, (▲) Ni, (▼) NiO, (✴) Li_0.1_Ni_0.9_O.

**Table 1 materials-13-04659-t001:** Corrosion rates of selected materials in molten NaOH in mm year^−1^.

Material	Temperature [K]								Time [h]	Reference
	673	693	773	783	853	873	953	973		
**Ni-201**	0.023	-	0.033	-	0.06	-	0.96	-	-	[14]
**Nickel**	-	-	0.195	-	-	-	-	-	288	[14]
**Copper**	13.4	-	17.2	-	-	-	-	-	288	[15]
**Inconel**	-	0.104	0.363	-	-	1.06	-	-	288	[15]
**Monel 400**	-	-	0.297	-	-	1.31	-	-	288	[15]
**Monel 500**	-	0.087	0.254	-	-	0.899	-	5.06	288	[15]
**Monel 600**	0.028	-	0.06	-	0.13	-	1.69	-	-	[14]
**301 SS**	0.043	-	0.08	-	0.26	-	1.03	-	-	[14]
**Carbon steel**	5.25	-	3.23	-	-	4.15	-	-	288	[15]
**Ni-Fe alloy**	-	-	-	1.8	-	-	-	-	336	[14]
**Austenitic steel, type 1**	-	-	-	15.9	-	-	-	-	336	[14]
**Austenitic steel, type 2**	-	-	-	24.2	-	-	-	-	336	[14]
**Ductile austenitic steel, type 2**	-	-	-	11.8	-	-	-	-	336	[14]
**Austenitic steel, type 3**	-	-	-	13.6	-	-	-	-	336	[14]
**Forged nickel**	-	-	-	0.23	-	-	-	-	336	[16]

**Table 2 materials-13-04659-t002:** Composition of St3SU carbon steel used in the construction of Model I MH-DCFC (wt.%).

C	Ni	Fe	C	Mn	S	Si	P
<0.22	0.3	Bal.	<0.24	<1.1	<0.05	0.10–0.35	<0.05

**Table 3 materials-13-04659-t003:** Composition of stainless steels used in the construction of Model II MH-DCFC (wt.%).

Material	Elemental Composition (wt.%)								
	Ni	Cr	Fe	C	Mn	S	Si	P	Other
**AISI 304L**	8–12	18–20	Bal.	<0.03	<2.0	<0.03	<1.0	<0.045	-
**AISI 316L**	10–14	16–18	Bal.	<0.03	<2.0	<0.03	<1.0	<0.045	Mo = 2.0–3.0
**AISI 316Ti**	10–14	16–18	Bal.	<0.08	<2.0	<0.03	<0.75	<0.045	Mo = 2.0–3.0 Ti = 0.7

**Table 4 materials-13-04659-t004:** Composition of nickel alloys used in the construction of Model III MH-DCFC (wt.%).

Material	Elemental Composition (wt.%)								
	Ni	Cr	Fe	C	Mn	S	Si	P	Other
**Nickel 201**	Min. 99	-	<0.4	<0.02	<0.35	<0.01	<0.2	-	Ti < 0.1 Cu < 0.25
**Inconel 600**	Min. 72	14.0–17.0	6.0–10.0	<0.15	<1.0	<0.015	<0.5	-	Cu < 0.5

**Table 5 materials-13-04659-t005:** The main parameters of the fuels used for MH-DCFC tests (all values are given for a “dry” state).

Fuel Sample	Elemental Composition (wt.%)					Ash (wt.%)	Volatile Matter (wt.%)	Higher Heating Value (MJ kg^−1^)
	C	H	N	S	O			
**Graphite rod**	97.4	0.6	0.2	0.00	0.10	1.7	1.2	32.35
**Commercial charcoal**	84.2	3.3	0.5	0.01	9.29	2.7	16.4	29.93

**Table 6 materials-13-04659-t006:** Changes in the mass of the high-nickel Inconel 600 alloy depending on the Model III operating time.

Time [h]	Mass Changes	
	[g]	[%]
**24 (lithiation process)**	+1.95	+6.1
**34 (first run after lithiation process)**	−0.0004	−0.0008
**77 (after 43 h long-term test)**	+0.0072	+0.0723

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
