# Peer review of "Materials Selection and Construction Development for Ensuring the Availability and Durability of the Molten Hydroxide Electrolyte Direct Carbon Fuel Cell (MH-MCFC)"

_materials, 2020, doi:10.3390/ma13204659_

Round 1

Reviewer 1 Report

Author designed and selected the materials for MH-DCFCs and studied their performance and behavior. Author clearly explained each condition for every performed experiment. They have shown the mass change or formation of corrosion product after the operation of the fuel cell. Following comments needs to be answered before its acceptance:

  1. Why do author used two different material for electrolyte tank in model II when there is a degradation at the contact point? Why nickel 201 was used for lower part? It was little confusing to comprehend for model II and model III. Provide the cost comparison of materials used for this set up.
  2. Model II was made from SS304L, SS316L and SS316Ti and were exposed to the temperature range of 773-873 K. Although these steels are not susceptible to intragranular corrosion, the long exposure between 700-1089K can still be harmful for low carbon SS. How can author address this issue when selecting a material for this temperature range?
  3. Why does author used only biochar as a fuel for long term experiment? How do they compare this test with graphite rod as a fuel? Please provide polarization data for these materials.
  4. Author tabulated the mass change for Inconel 600 to observe the degradation. What is the mass change of the materials used in model I and model II after the performance? Please provide data using electrochemical technique to monitor the fluctuations in current and potential during the operation of a fuel cell? Also, caption for table 6 is confusing, please make it clear. Is it model II or model III? Inconel 600 was used for model III.
  5. Please proofread the document to avoid any typo error in the text.
  6. See line 174, 210, 214, 327, 577 to make it clear.

Author Response

  1. Why do author used two different material for electrolyte tank in model II when there is a degradation at the contact point? Why nickel 201 was used for lower part? It was little confusing to comprehend for model II and model III. Provide the cost comparison of materials used for this set up.

The intention of the authors of using nickel 201 as one of the elements of the electrolyte reservoir was to initially determine the suitability of this material as a target material for the construction of the cell components.

Nickel 201 is a relatively expensive material compared to stainless steels. On the other hand, the energy and economic analysis will be the subject of another article planned by the authors.

  1. Model II was made from SS304L, SS316L and SS316Ti and were exposed to the temperature range of 773-873 K. Although these steels are not susceptible to intragranular corrosion, the long exposure between 700-1089K can still be harmful for low carbon SS. How can author address this issue when selecting a material for this temperature range?

The comments made by the reviewer regarding the corrosion resistance of SS steel were predicted, which is shown in the results of the research obtained by the authors. As a result, this type of cell material was eliminated and finally nickel-based materials were indicated.

In addition, it should be noted that the tested MH-DCFC cell operates at temperatures up to 723 K, therefore, taking into account the range indicated by the reviewer, the risk of degrading the effects of very high temperature is minimal.

  1. Why does author used only biochar as a fuel for long term experiment? How do they compare this test with graphite rod as a fuel? Please provide polarization data for these materials.

The tested DCFC cell is intended to work with fragmented fuels such as hard coal, biochar, activated carbons, etc. (fuel grains should be placed in a special anode structure, which made it possible to study the influence of the corrosive environment on this element of the cell structure. In the case of graphite, it is not necessary to use steel as anode elements).

Moreover, after the long-term test with the use of biochar, it was not possible to conduct another such test with the use of graphite due to the degradation of the cell's structural materials, which is illustrated by the test results described in the article.

  1. Author tabulated the mass change for Inconel 600 to observe the degradation. What is the mass change of the materials used in model I and model II after the performance?

The changes in the mass of materials used in Models I and II were not made because the degradation and corrosion of the materials was so clear that the focus was only on the microstructural and phase analysis and, above all, on the analysis of the cell's operation.

  1. Please provide data using electrochemical technique to monitor the fluctuations in current and potential during the operation of a fuel cell?

The reviewers’ comment was taken into consideration and more specific information were added to the “2.5. MH-DCFC performance tests” section (cf. red-marked text in revised manuscript).

  1. Also, caption for table 6 is confusing, please make it clear. Is it model II or model III? Inconel 600 was used for model III.

The caption for Table 6 was confusing and taking into consideration the reviewers’ comment the caption was modified.

  1. Please proofread the document to avoid any typo error in the text. See line 174, 210, 214, 327, 577 to make it clear.

The manuscript was carefully checked and corrections were incorporated into the text (cf. red-marked text in revised manuscript).

Reviewer 2 Report

  1. The introduction part should be elaborated.
  2. Authors should rewrite the abstract. It includes the important results of this work.
  3. English language of the manuscript should be polished.
  4. Did all the digital photographs given by the authors original?
  5. Authors should provide a comparison table to compare the results of present work with other reported literature.

Author Response

1.The introduction part should be elaborated.

The reviewer’s comment was taken into consideration and the “introduction” section was revised and modified.

2.Authors should rewrite the abstract. It includes the important results of this work.

The reviewer’s comment was taken into consideration and to make things clear the abstract was modified.

3. English language of the manuscript should be polished.

The manuscript was carefully checked and corrections were incorporated into the text (cf. red-marked text in revised manuscript).

4.Did all the digital photographs given by the authors original?

Yes. 

5. Authors should provide a comparison table to compare the results of present work with other reported literature.

To made things clear we have added more specific information (cf. page 14, lines 441-448 and page 19 lines 577-582).

Reviewer 3 Report

No

Author Response

Thak you for your revision.

Reviewer 4 Report

 The manuscript describes the corrosion of different materials in the models of molten hydroxyde fuel cell. The topic is important for the community, and I recommend to publish the manuscript after revision. The Materials and methods section of the manuscript looks to long. Table 1 should be moved to the introduction with relevant discussion, concerning the material selection for fuel cell prototypes, because it is not describing experimantal details, but literature review. Authors may consider shortening of the remaining part of  M&M section, with moving some technical details to the supplementary information. Results section looks relevant and needs minor typo corrections like upper index for "10-4" at p7 l216. To make images more clear, it is better to use colored arrows with the same color as a data plot in Figs. 12b, 16b, 20b to help to understand the axis relevant for each curve. Red curves for power and current on Figs. 12b and 20b are visually connected together and mislead the reader, some manipulations with the scales are needed to make them look more clear.   

Author Response

1. The Materials and methods section of the manuscript looks to long. Table 1 should be moved to the introduction with relevant discussion, concerning the material selection for fuel cell prototypes, because it is not describing experimantal details, but literature review.

The reviewer’s comment was taken into consideration and the Table 1 was moved to “introduction” section.

 2. Authors may consider shortening of the remaining part of  M&M section, with moving some technical details to the supplementary information.

According to the authors, moving parts of the description or photos from the M&M chapter will make the manuscript less readable for the reader. Extensive part of the information from the M&M chapter has been moved to the Introduction as recommended by the reviewer

3. Results section looks relevant and needs minor typo corrections like upper index for "10-4" at p7 l216.

The manuscript was carefully checked and corrections were incorporated into the text

4. To make images more clear, it is better to use colored arrows with the same color as a data plot in Figs. 12b, 16b, 20b to help to understand the axis relevant for each curve.

Red curves for power and current on Figs. 12b and 20b are visually connected together and mislead the reader, some manipulations with the scales are needed to make them look more clear.   

Figs. 12b, 16b, 20b were modified based on the suggestion of the Reviewer.

Round 2

Reviewer 1 Report

Author commented on all the issues that were raised and have made corresponding changes in the manuscript. The manuscript can be accepted in present form.